# Automatic Monitoring Alarm Method of Dammed Lake Based on Hybrid Segmentation Algorithm

**DOI:** 10.3390/s23104714

**Published:** 2023-05-12

**Authors:** Ziming Cai, Liang Sun, Baosheng An, Xin Zhong, Wei Yang, Zhongyan Wang, Yan Zhou, Feng Zhan, Xinwei Wang

**Affiliations:** 1School of Resources, Environment and Materials, Guangxi University, Nanning 530004, China; 2Optoelectronic System Laboratory, Institute of Semiconductors, Chinese Academy of Sciences, Beijing 100083, China; 3Institute of Tibetan Plateau Research, Chinese Academy of Sciences, Beijing 100101, China; 4School of Electronic, Electrical and Communication Engineering, University of Chinese Academy of Sciences, Beijing 100049, China

**Keywords:** dammed lake, automatic monitoring alarm, k-means clustering algorithm, region growing algorithm, water level recognition

## Abstract

Mountainous regions are prone to dammed lake disasters due to their rough topography, scant vegetation, and high summer rainfall. By measuring water level variation, monitoring systems can detect dammed lake events when mudslides block rivers or boost water level. Therefore, an automatic monitoring alarm method based on a hybrid segmentation algorithm is proposed. The algorithm uses the k-means clustering algorithm to segment the picture scene in the RGB color space and the region growing algorithm on the image green channel to select the river target from the segmented scene. The pixel water level variation is used to trigger an alarm for the dammed lake event after the water level has been retrieved. In the Yarlung Tsangpo River basin of the Tibet Autonomous Region of China, the proposed automatic lake monitoring system was installed. We pick up data from April to November 2021, during which the river experienced low, high, and low water levels. Unlike conventional region growing algorithms, the algorithm does not rely on engineering knowledge to pick seed point parameters. Using our method, the accuracy rate is 89.29% and the miss rate is 11.76%, which is 29.12% higher and 17.65% lower than the traditional region growing algorithm, respectively. The monitoring results indicate that the proposed method is a highly adaptable and accurate unmanned dammed lake monitoring system.

## 1. Introduction

Global warming causes the occurrence of extreme weather and the rapid melting of glaciers, affecting the environment and the safety of humans [1,2]. As the world’s third pole, the Qinghai-Tibet Plateau is more sensitive to climate change and has a more delicate ecosystem [3,4,5,6]. In October 2018, the Grand Canyon on the Yarlung Tsangpo River (YTR) in the Sedongpu Basin experienced two successive glacier collapses and river-blocking incidents, and they caused a huge disaster and posed a potential threat to the residents and transport lines upstream and downstream from Paizhen Town and in the area along the river banks in Medog County. The incidents produced a disaster chain process of glacier collapse, glacial debris flow, river blockage, dammed lake, and outburst flood [7]. With the climate warming on the Tibetan Plateau, such disasters will continue or even intensify in the future. To ensure the safety of human lives and property and to better comprehend the law of glacier movement, it is of considerable practical importance to conduct research on the automatic dammed lake event alarm.

It is feasible to detect whether a dammed lake event is caused by observing the river water level because variations in water level are typically associated with river obstacles. Currently, numerous efforts have been made to monitor the variation in water level, such as pressure sensors [8,9,10,11], ultrasonic meters [12,13,14,15,16], and optical vision monitoring [17,18]. Pressure sensors must be calibrated, and they are extremely sensitive to any lateral movement at mounting points. Continuous water pressure can cause sensor breakdown, requiring frequent calibration and replacement. Ultrasonic sensors use ultrasonic pulses to measure the time it takes for a signal to travel from an emitter to a receiver, but their lifetime is short and the returned sound wave values are sensitive to temperature, precipitation, and snowfall. Optical image monitoring applications are affordable and provide a vast monitoring area, compared with conventional fixed-point sensors, to detect water levels [19]. In terms of system maintenance, the maintenance of pressure sensors and ultrasonic meters is frequently a time-consuming and resource-intensive endeavor, especially for the disaster monitoring devices installed in uninhabited areas, due to a variety of environmental and technical reasons. Optical imaging monitoring is stable, but it generates a large number of images that require manual judgment and consumes a lot of manpower. In order to save costs and liberate manpower, the realization of automatic monitoring has become an urgent problem.

Ground-based monitoring systems are well suited for long-term fixed-point automatic monitoring of specific small areas, such as wild rivers, urban rivers, drainage ditches, and so on. Optical satellite and aerial monitoring systems are often used for large-scale geographical analysis because their accuracy is not as high as that of ground monitoring due to long observation ranges, and their monitoring of fixed sites is not continuous. Water level identification methods for automatic monitoring are classified into two types based on whether or not manual marking is used. Using artificial markers, such as a water gauge, can determine the variation in water level, and recognizing the variation in the river boundary also can determine the variation in water level. When the monitoring site has a water gauge marker, the intensity information on both sides of the water level will be different, and the variation of the water level can be obtained by extracting the reflection parameter of the river water bodies from the image [20,21]. In addition, when the image template is extracted from the marker in advance, the part of the template that can be matched by the multi-template matching technology is the area that has not been covered by the water, allowing the variation in water level to be obtained. [22,23]. The manual placement of water gauges is highly limited because it is often inaccessible at many monitoring sites. However, automatic monitoring has grown in popularity [24,25,26,27] and the key to achieving automatic monitoring is the variation in water level. When there is no gauge marker, the variation in river boundary can provide information on the variation in water level. So, the extraction of the river boundary is critical for automatic water level identification.

Canny edge detection is a simple and effective image processing technology proposed for application in real-world environments [28,29]. However, the bank lines of natural rivers without artificial embankments are fuzzy, and canny edge detection cannot identify lines precisely. The region growing (RG) algorithm is often used to identify the region of water due to the uniformity of watercolor in favor of seed growth. The hybrid algorithm achieves relatively higher precision in flood area identification compared with growth cutting and RG alone [30]. Other hybrid algorithms can also identify flood zone with relatively high levels of accuracy, such as threshold value and RG [31]. To achieve a good identification effect in a complex river scene, the above algorithms require regular parameter modification, which is not conducive to automatic monitoring. Therefore, this paper proposes a hybrid segmentation algorithm for automatic monitoring alarm systems in no man’s land that does not rely on artificial markers.

The remainder of this paper is organized as follows. Section 2 describes the automatic monitoring alarm system and hybrid segmentation algorithm in detail. Section 3 shows the experimental results of the proposed method, and Section 4 gives conclusions.

## 2. Methodology

### 2.1. Automatic Monitoring Alarm System

Lhasa Earth System Multi-Dimension Observatory Network established a long-term automatic monitoring and early warning system focused on glacier collapse disaster chains in the Grand Canyon on the YTR [7]. In the system, the automatic monitoring alarm sub-system (AMAS) is an image-based unattended monitoring unit that decides whether a dammed lake event has occurred at a monitoring site. The AMAS analyzes variations in water level by combining images captured on-site with a back-end image processing module. When the water level rises, the system sends an alarm indicating that the lake has been dammed. Figure 1 shows that the system consists of a camera, satellite, and back-end image processing module. Images are transmitted by satellite to the back-end image processing module, and the proposed algorithm is used to detect the water level. When an alarm happens, the user will receive a short messaging service (SMS) that a dammed lake has occurred.

### 2.2. Hybrid Segmentation Algorithm

The proposed algorithm is a combination of RG and k-means clustering algorithms. The goal of RG is to separate the greenish river pixels, and the goal of k-means clustering is to classify all pixels, so green channel gray image and full-channel gray image are used, respectively. The input RGB images are divided into two types of gray images. The gray value of full-channel gray images adopts 30% red channel, 59% green channel, and 11% blue channel. The gray value of green channel gray images adopts 100% green channel. Step 1 shows the process of selecting the region of interest (ROI) and the detection line. Step 2 shows the process of using the RG algorithm to obtain the RG line as the auxiliary line on the green channel of an image. Step 3 shows the process of using the k-means clustering algorithm on denoised images. Step 4 shows the process of obtaining water level by combining the skeleton thinning algorithm based on k-means clustering with the RG-line. The region of interest (ROI) is a manually selected rectangular area of the image. Step 5 shows the process of making alarm judgments by using pixel water level data of the waterline.

The proposed method’s flow diagram is shown below, Figure 2b excludes include step 5, which is the alarm strategy.

Step 1:

Region of interest (ROI) is selected to avoid unneeded information interference. It reduces the influence of low-resolution regions and improves the performance of image segmentation. The detection line gives additional scene information helping the RG algorithm to segment the scene.

The details of the processing are shown in Figure 2 based on an image from the YTR on 24 April 2021. The a priori knowledge image in Figure 2b shows the ROI and detection line selection. Due to the depth of field span of the image being large, we select the right 1/3 area as the monitoring ROI for better resolution of river boundary and larger river area.

The water level is low in April and high in August according to the hydrological characteristics of the central basin of the YTR. As shown in the a priori knowledge image in Figure 2b, we positioned the detection line at the position of the yellow line where the water level is believed to be unreachable. In August, the detection line was personally confirmed. When the water level exceeds the detection line, move the water level up. When the detection line is not exceeded by the water level, keep it unchanged.

Step 2:

RG algorithm is the process of aggregating pixels or subregions into larger areas according to predefined criteria. The location of the seed point and the growth criterion are the parameters involved in the RG algorithm. The algorithm evaluates the relationship between each seed point and its eight neighboring areas. Similar points are used as the seed points for the next growth. The algorithm stops growing when no similar seed points exist.
(1)$$p\left(x_{1}{,y}_{1}\right)-p\left(x_{2}{,y}_{2}\right)>G$$
where, $p\left(x_{2}{,y}_{2}\right)$ is the pixel gray value of seed points, $p\left(x_{1}{,y}_{1}\right)$ is the gray value of adjacent pixel points, and G is the gray difference threshold.

In this study, the location of the seed point is obtained by automatically spreading seed points in a square grid. Each seed point produces an RG image. Overlay all RG images that do not overlap the detection line. The max-connected domain is derived from the overlay graph, while RG-line is derived from the water level-side boundary information of the max-connected domain graph. The growth criterion is the gray value difference on both sides of the water level. The RG image in Figure 2b shows the maximum connectivity domain and adjunction region. The average pixel width of the river is roughly estimated as the side length of the square grid and half the width of the adjunction region.

Step 3:

K-means clustering is a classification algorithm that classifies pixels in an image into K classes, while the Non-local mean (NLM) algorithm is used to obtain the denoised image. The K-mean clustering algorithm is achieved by iteratively updating the initial estimate of the class center, using the procedures below:
(1)Initialize class gravity center $\mu_{i}$ with i=1⋯n;(2)Assign each pixel to the nearest center of class C;(3)The update center is the mean of the gray value of pixels that are assigned to a particular class;(4)Repeat (2) and (3) until the algorithm converges.

The K-means algorithm minimizes as much as possible the variance V between classes. The variance V is given by
(2)$$V=\sum_{i=1}^{k}\sum_{x_{j}\in C_{i}}{\left(x_{j}-\mu_{i}\right)}^{2}$$
where, $x_{j}$ represents the jth pixel, 1≤j≤n. $\mu_{i}$ represents the mean of the ith cluster, 1≤i≤m. $C_{i}$ represents the ith cluster.

The K-means clustering image in Figure 2b shows the result of the clustering of all pixels in ROI. ROI has three primary objectives including the river and both sides of the river. The parameter K in the k-means clustering algorithm is related to the number of primary targets. So, we select K = 4. As a transition category, a new category has been added.

Step 4:

The principle of the traditional skeleton thinning algorithm is to extract the center contour of the object on the image. The algorithm uses 3 × 3 pixel windows to thin out the target from the target boundary to the target center until it is corroded to the point where it cannot be corroded (width of a single-layer pixel) in binary images. Then the image skeleton is obtained.

Our proposed boundary-based skeleton thinning algorithm is to extract the boundary skeleton instead of the center skeleton. Image skeleton extraction based on the boundary is divided into two steps: (1) Extract the contour of the target and record these contour points. (2) Detect whether x pixels below these contour points are all non-target pixels in turn (in a binary image, the non-target pixel is represented by a color difference with the contour point pixel). When all pixels are non-target pixels, the contour point is a river boundary point pixel. When all pixels are not non-target pixels, the contour point is not a river boundary point pixel. Where, x is related to the estimated average river width, generally one-tenth of the average river width is appropriate. We use the RG-line as a supplementary line to help remove noisy pixels from the skeleton thinning graph. The waterline is derived from the image boundary of the processed skeleton thinning graph on the water level side.

The intermediate process image in Figure 2b shows the position of the adjunction region in the k-means clustering graph.

The skeleton pixel image in Figure 2b shows the result of thinning skeleton of four types of pixels (red, green, blue, and yellow) and the process of filtering in the adjunction region of the skeleton graph. The skeleton graph has a linear shape, and the number of pixels within a connected component reflects the continuity of this boundary. Firstly, compare the size of the number of pixels in the maximum connected components for each pixel type. Secondly, compare the size of the number of pixels in the second connected component. Finally, compare the size of the number of pixels in the third connected component. In each comparison, the pixel type with the most pixels gets one point. When the score is the same, the pixel type with more pixels in the maximum connected component is preferred.

The blue pixel image in Figure 2b shows that the selected pixel categories are those with the highest score.

The resulting image in Figure 2b shows the process of removing all pixels below the dividing line by taking the RG line as the dividing line to obtain the waterline.

Step 5:

The monitoring system analyzes the difference in pixel water level to determine whether a dammed lake event has happened. The user set the water level alarm threshold, and the system alarm when the difference exceeds the threshold. The water level threshold used in this paper is 2 pixels. The pixel water level is the number of pixels within the image at the water level height under a fixed field of view. The difference in pixel water level equates to the difference in water level between two images captured at adjacent monitoring times.

Figure 3 shows the analysis of the difference in water level. Figure 3a shows the statistical differences between the pixel water level on 1 May 2021 and 24 April 2021. Figure 3b shows a magnified image of the top three peaks. The value of pixel water level difference of −1, 0, and 1 is regarded as the numerical fluctuation caused by algorithm resolution, so these values are not considered. In the top three peaks, when the number of negative values exceeds the number of positive values, the water level is regarded as rising, and when the number of positive values exceeds the number of negative values, the water level is regarded as falling. Considering the possibility of no variation in water level, a ratio of 0 over 50% is considered as reflecting no variation in water level.

## 3. Results and Discussion

### 3.1. Dataset

The proposed AMAS was used in Nyingchi, Yarlung Tsangpo River Basin, Tibet Autonomous Region, China. The river is frequently blocked by glacial debris flows, and causing a significant risk of the dammed lake. The format of the image is png, and the capture rate of the image is one frame per hour. The AMAS software interface consists of reference images, input images, and parameter settings. The reference image is the initial image of the water level, and the input image is the image that needs to be evaluated. When water levels exceed a predetermined threshold, the AMAS shows an alarm screen.

The river region is saturated with water vapor and sunlight, providing a stable temperature inversion. Hence, fog is common in the morning. Due to the viewing angle, the camera tends to produce overexposed images during midday. An hour after noon, when the sunlight shifts to the right angle is a good window for observation. Notably, noon in Tibet is two to three hours later than Beijing time because of the difference in longitude. In conclusion, we choose 4 pm as the viewing window. Figure 4 shows the clear image, the halo image, and the fog image from left to right.

### 3.2. Results of the Proposed Scheme

In order to estimate the boundary recognition performance, the images identified by the proposed algorithm were compared with the RG algorithm. The results show that the proposed algorithm has a higher accurate ability for recognizing boundaries than the RG algorithm.

Figure 5 shows the statistical distribution of all image data collected during the test. The y-coordinate shows the time change over the course of a day, and the x-coordinate shows the time change over the course of a month. The vacant position indicates that the camera lacks power, resulting in failure to take images. The red position indicates that the target river is obscured by a hole or fog in the image. The green position indicates a normal photograph. The blue box is the viewing window at 16:00.

Figure 6 shows the monthly variations in the water level identified by the proposed algorithm. In the selected area, the black lines represent the boundary lines identified by our method, and the red dots are the transverse centers of the boundary lines. The fluctuating water level follows a pattern of initially rising, then lowering and then rising, with the highest water level occurring in August.

Significant gray differences created by white water waves will have a negative effect on the RG algorithm, which uses gray differences as a growth criterion, causing the algorithm to terminate growth before the region of the target is completely detected. In addition, noise interference still presents in the image after denoising, resulting in the identification of the non-river region as a river region. K-means clustering employs Euclidean distance based on pixel gray value as the judging criteria to classify pixels across the whole image. Compared to the local gray difference comparison of the RG algorithm, the global gray difference comparison is able to distinguish between the regions of white water waves and rivers more accurately. The proposed method is able to recognize river border pixels accurately. Figure 7 shows the visual comparison results of different methods.

Figure 7a,b and c, respectively, show the enlarged visual comparison of the water level on 24 April 2021, 1 May 2021, and 9 May 2021. Figure 7d shows a visual comparison of water levels from 17 May 2021 to 24 November 2021. The boundary pixels in the contrast image underwent a 5 × 5 convolution core operation for better visual presentation. The accuracy is the proportion of samples correctly identified by the classifier out of the total number of samples. Although accuracy can evaluate the overall correctness of a classifier, it is not a very efficient measuring indicator when the categories in the total samples have a skewness distribution. Even if the classifier’s classification accuracy is greater than 99%, this does not necessarily indicate that the model is good. Hence, the classifier will frequently use the accuracy rate and recall rate to measure. We use the binary model index to compare the alarm methods using different segmentation algorithms. The indexes of the binary model are accuracy (ACC), precision (PRE), recall (TPR, True Positive Rate), false alarm (FPR, False Positive Rate), and miss rate (FNR, True Negative Rate).
(3)ACC=(TP+TN)/(TP+FN+FP+TN)
(4)PRC=TP/(TP+FP)
(5)TPR=TP(TP+FP)
(6)FPR=FP(FP+TN)
(7)FNR=FN(TP+FN)
where, True positive (TP) means that the positive sample is predicted to be positive, False negative (FN) means that a positive sample is predicted to be negative, False positive (FP) means that a negative sample is predicted to be positive, and True negative (TN) means that a negative sample is predicted to be negative. The results show that the proposed algorithm is superior to the traditional RG algorithm in accuracy, precision, recall, false alarm, and miss rate.

When all samples were evaluated to be negative, the accuracy index remained extremely high if the majority of samples were negative. The precision index suffers from the same issue, when a False positive (FP) is low, the precision index will be high. Therefore, we measured the recall (TPR) of statistical results. Figure 8 shows the statistics and analysis of dammed lake alarm. Figure 8a shows the statistics of the alarm. Figure 8b shows the comparison of indicators using different methods. The proposed algorithm improved in these three dimensions by 29.12%, 27.08%, and 17.65%, respectively. The proposed algorithm decreases false alarm and miss rates by 45.46% and 17.65%, respectively. The proposed algorithm is superior to the RG algorithm in all five dimensions.

## 4. Conclusions

In this paper, a hybrid algorithm based on RG and k-means clustering is proposed for an automatic monitoring alarm sub-system for dammed lake events. Firstly, ROI and detection lines are used to provide prior knowledge of the scenario. Then, a hybrid segmentation algorithm is proposed to identify river boundaries. The results of the visual comparison show that the proposed algorithm has more accurate boundary recognition ability than the RG algorithm. Lastly, the identified boundaries are analyzed numerically. The results show that the proposed algorithm performs well in quantitative measurement compared with other methods. Regarding accurate alarms, the performance of the proposed method is better than that of the RG method in accuracy (89.29%), precision (93.75%), and recall (88.23%). In terms of its ability to reduce errors, the proposed method is also superior to the RG method in false alarm (9.09%) and miss rate (11.76%), hence decreasing resource waste caused by false alarms. In the future, the proposed method will be tested in different geographical regions to strengthen the algorithm’s robustness, and at the same time, image enhancement will be carried out in image preprocessing to filter halo and fog and obtain clear images to realize better performance of the proposed method. This research proposes an automatic monitoring alarm sub-system as a solution to the problem of dammed lake monitoring. Future applications of this method include flood warnings and seasonal river state judgment.

## Figures and Tables

**Figure 1 sensors-23-04714-f001:**
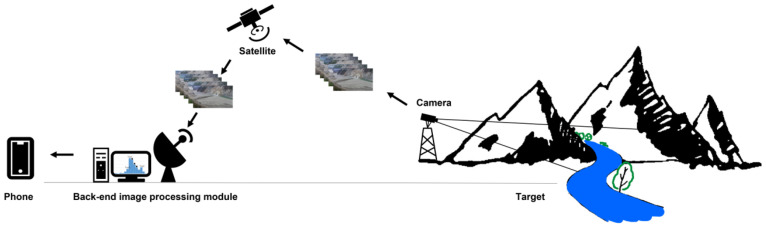
Automatic monitoring alarm sub-system for dammed lake.

**Figure 2 sensors-23-04714-f002:**
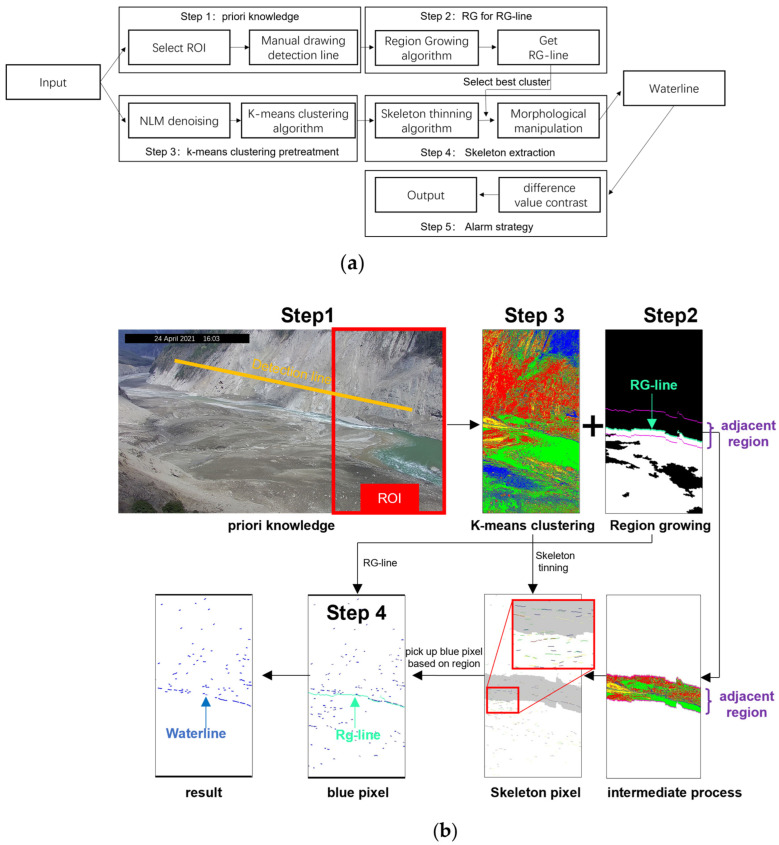
(**a**) Flow diagram of the proposed method. (**b**) Flow chart of the proposed algorithm using the real data on 24 April 2021 from the YTR. The color is used to distinguish between different types of pixels. The pixels of the image have been divided into 4 categories using K-means clustering. The red box of the priori knowledge image is a region of interest. The red box of skeleton pixel image is an enlarged image of the area.

**Figure 3 sensors-23-04714-f003:**
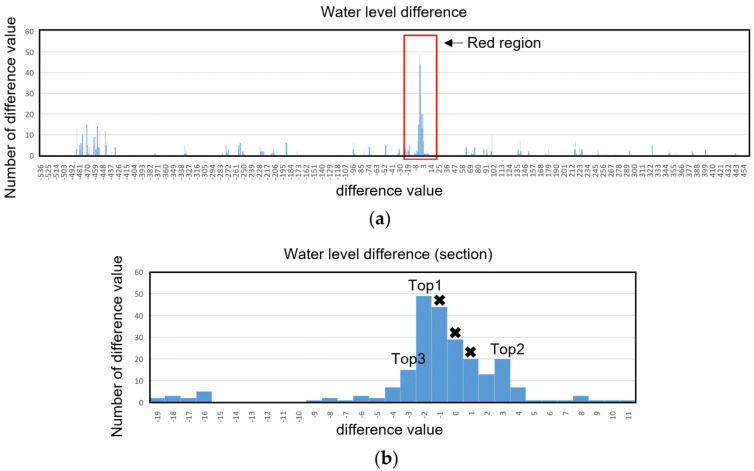
Difference in water level analysis process. The waterline of 24 April 2021 and 1 May 2021 is used for example analysis. (**a**) Statistical results of difference in pixel water level at different positions. (**b**) Enlarged graph of the difference in pixel water level at different positions of the top three peaks in the red region. A cross indicates that the data there has been removed due to numerical fluctuation.

**Figure 4 sensors-23-04714-f004:**
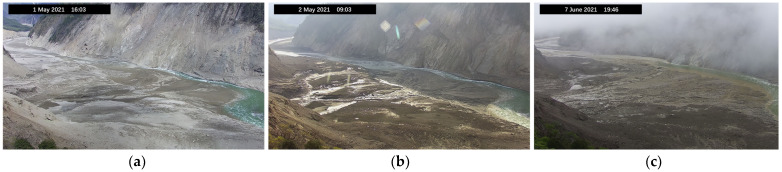
Pictures of monitoring scenarios. (**a**) Image that can be recognized by the algorithm. (**b**) and (**c**) Occluded Image that were not recognized by the algorithm due to halo or fog.

**Figure 5 sensors-23-04714-f005:**
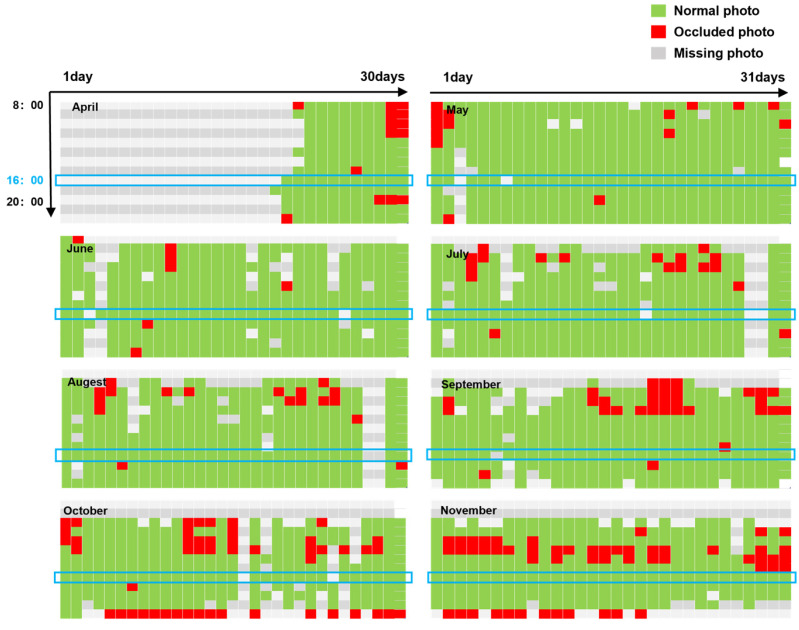
Distribution of observed data across time from April to November. The observation window is in the blue box.

**Figure 6 sensors-23-04714-f006:**
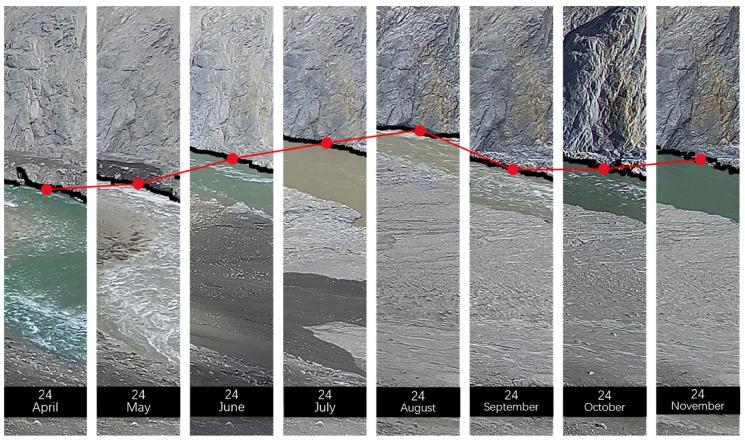
Monthly variation of water level from 24 April 2021 to 24 November 2021. The red dots are the transverse centers of the boundary lines and Red lines make visual changes easier to notice.

**Figure 7 sensors-23-04714-f007:**
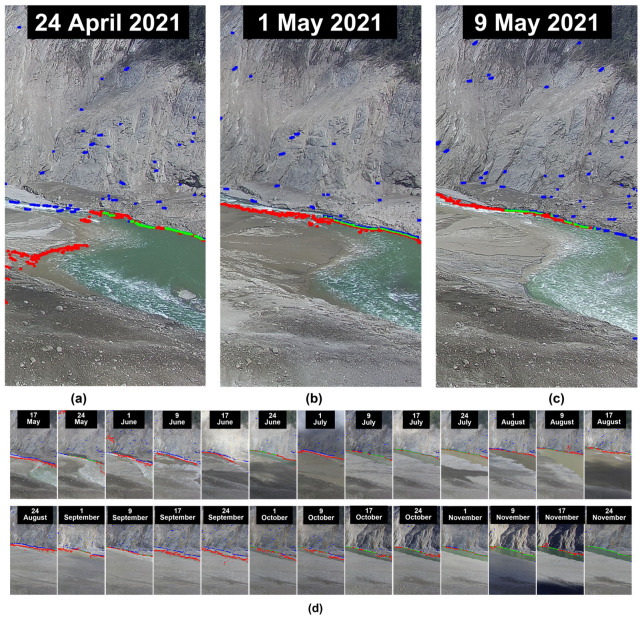
Visual comparison. Observations are made every seven days from 24 April 2021 to 24 November 2021, and the color red denotes the water level identified by RG. The color blue is the water level identified by the proposed algorithm. The color green indicates the overlap of water level identified by the two algorithms. (**a**) Visual comparison of 24 April 2021 (**b**) Visual comparison of 1 May 2021 (**c**) Visual comparison of 1 May 2021 (**d**) Visual comparison of 1 May 2021 to 24 November 2021.

**Figure 8 sensors-23-04714-f008:**
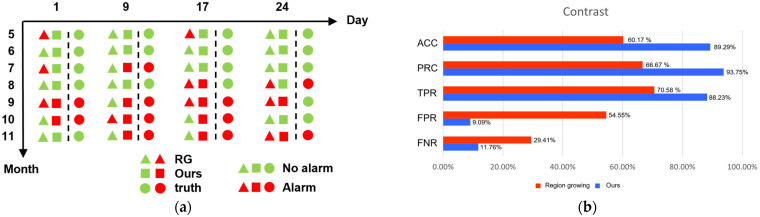
Alarm situation and indicator comparison. We counted the alarms from 1 May 2021 to 24 November 2021 and compared the corresponding indicators. (**a**) Alarm distribution. Green indicates that the system does not automatically alarm, and red indicates that the system automatically alarms. The three cells in a monitor, from left to right, are labeled as follows: RG, Ours, and truth. For simpler visual comparison, the true values are divided by a dotted line. (**b**) Indicator comparison. Comparison between RG and Ours in ACC, PRE, TPR, FPR, and FNR.

## Data Availability

The data is unavailable due to privacy.

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
