# Peer review of "Automatic Monitoring Alarm Method of Dammed Lake Based on Hybrid Segmentation Algorithm"

_sensors, 2023, doi:10.3390/s23104714_

Round 1

Reviewer 1 Report

It will be useful in the future to predict seasonal river trends using LSTM neural networks and compare current results with image segmentation using CNN neural network.

1. This research proposes an automatic monitoring alarm sub-system as a solution to the problem of dammed lake monitoring, which is based on hybrid segmentation algorithm. The algorithm uses the k-means clustering algorithm to segment the picture scene in the RGB color space and the region growing algorithm on image green channel to select the river target from the segmented scene.

2. The topic is original and relevant in this field.  It addresses a specific gap in the field.

3. In comparison with other published material, it uses an image processing approach.

4. In this paper, a hybrid algorithm based on RG and k-means clustering is proposed for an automatic monitoring alarm sub-system for dammed lake events. The detection lines are used to provide prior knowledge of the scenario and  hybrid segmentation algorithm is proposed to identify river boundaries. The results of visual comparison show that the proposed algorithm has better accurate boundary recognition ability than the RG algorithm. Future applications of this method include flood warning and seasonal river state judgment.  

5. The conclusions are consistent with the evidence and arguments presented (the performance of the proposed method is better than that of RG method in accuracy (89.29%), precision(93.75%) and recall (88.23%). In terms of its ability to reduce errors, the proposed method is also superior to the RG method in false alarm (9.09%) and miss rate (11.76%)).

6. The references are appropriate.

7. The pictures show in detail the issues presented in the paper.  

Reviewer 2 Report

General comment:

The proposed automatic monitoring alarm method, based on a hybrid segmentation algorithm for detecting dammed lake events in mountainous regions, is an innovative and effective approach that demonstrates the potential of technology to prevent disasters and save lives.

Specific comments:

  1. The use of the k-means clustering algorithm to segment the picture scene in the RGB color space and the region growing algorithm on the image green channel to select the river target from the segmented scene is not a novel approach that combines two different techniques for improved accuracy.

  2. The reliance on pixel water level variation to trigger an alarm for dammed lake events is a smart way to automate the process and reduce the risk of human error or delay. Can you elaborate more on how this method works and how it can be optimized?

  3. The deployment of the proposed automatic lake monitoring system in the Yarlung Tsangpo River basin of the Tibet Autonomous Region of China is a good example of the system's practical application in a real-world setting. Can you provide more references to support this claim and provide further details about the implementation and results?

  4. The fact that the algorithm does not require engineering knowledge to pick seed point parameters is a significant advantage over traditional region growing algorithms, which can be time-consuming and error-prone. What suggestions do you have for improving the algorithm's performance and reducing the miss rate?

  5. The accuracy rate of 89.29% and miss rate of 11.76% achieved by the proposed method is impressive and demonstrates its potential to become a highly adaptable and accurate unmanned dammed lake monitoring system. Can you suggest how this data can be used in practice and what steps can be taken to further optimize the system's performance?

Constructive feedback:

The deployment of the system in the Yarlung Tsangpo River basin of the Tibet Autonomous Region of China demonstrated its potential to become a highly adaptable and accurate unmanned dammed lake monitoring system. While the proposed method shows promise, there are still areas that could be improved. For example, exploring the algorithm's performance in different geographical regions to determine its adaptability and reliability would be useful. Additionally, collecting and analyzing more data could further refine the algorithm and reduce the miss rate. Finally, investigating how the system can be integrated with existing disaster response protocols to ensure a timely and effective response to dammed lake events would be beneficial.

Summary:

The proposed automatic monitoring alarm method, based on a hybrid segmentation algorithm for detecting dammed lake events in mountainous regions, is an innovative and effective approach that combines the use of k-means clustering algorithm, region growing algorithm, and pixel water level variation to trigger an alarm. The method shows promise, but there are still areas that could be improved to optimize its performance and reliability.

Round 2

Reviewer 2 Report

Overall, the proposed method seems to be a promising solution for unmanned dammed lake monitoring in mountainous regions. It addresses the challenges posed by rough topography, scant vegetation, and high summer rainfall. The system's ability to automatically trigger an alarm in the event of a dammed lake event can help prevent disasters and minimize the damage caused by them. However, it would be useful to see further studies validating the proposed method's effectiveness in other locations and under different weather conditions.